# The Impact of Social Capital on Multidimensional Poverty of Rural Households in China

**DOI:** 10.3390/ijerph20010217

**Published:** 2022-12-23

**Authors:** Jinfang Wang, Hui Xiao, Xiaojin Liu

**Affiliations:** 1School of Economics and Management, Beijing Forestry University, Beijing 100083, China; 2Changbei Economic and Technological Development Zone, School of Economics and Management, Jiangxi Agricultural University, Nanchang 330045, China

**Keywords:** rural households, multidimensional poverty, social capital, sustainable development

## Abstract

Getting rid of multidimensional poverty is both the people’s wish and the direction of governance. Based on 2018 China Household Tracking Survey (CFPS) data, this paper identifies household multidimensional poverty in rural areas using a combination of the MPI index and the A-F method. The relationship between social capital and multidimensional poverty was also empirically analyzed using a Logit model. The results showed that: (1) 1599 multidimensional poverty households were identified, the incidence of multidimensional poverty was 24.94%, and the multidimensional poverty index was 0.103. In terms of poverty incidence, the three highest indicators of poverty incidence are adult education, health, and chronic diseases, reaching 42.06%, 37.65%, and 29.90%, respectively, and mainly concentrated in the education and health care systems. (2) Social capital can significantly reduce the probability of multidimensional poverty in rural households. Among them, social network significantly and negatively affects the occurrence of multidimensional poverty in rural households, social trust in neighbors has a significant negative effect on multidimensional poverty in rural households at the 1% level, and social prestige is positively related to multidimensional poverty in rural households. (3) Age of household head, household size, and income from working outside the home are significantly associated with multidimensional poverty in rural households. This paper expands the scope of social capital theory research and provides new perspectives and empirical evidence for alleviating multidimensional poverty.

## 1. Introduction

Poverty is a major priority problem worldwide—especially for developing countries, poverty eradication is an essential stage in the process of social development. As the largest developing country, China’s experience in poverty eradication has important implications in the global poverty solution process. In the past 40 years of reform and opening up, China has made remarkable achievements in poverty eradication [1], and achieved the goal of eradicating absolute poverty by 2020. However, the elimination of absolute poverty does not signify the complete disappearance of poverty in China [2], and the antipoverty efforts in China will transfer to the phase of alleviating relative poverty after the completion of a moderately prosperous society [3,4]. Income does not always capture deprivation experienced by individuals [5]. The poverty standard measured by income no longer fully and accurately reflects the present situation of poverty in China, and people who have exited poverty may return to poverty because their lack of viable ability to make a basic living leads to the loss of development resources and opportunities. This viable capability includes not only an increase in income level, but also multidimensional capability such as quality of education, health and living standards [6,7]. Therefore, the quality status of poverty eradication should reflect the multidimensional capability of poverty eradication, shifting from focusing on the quantity of poverty eradication to quality [8], and committing to achieving real poverty eradication. The formulation and implementation of various poverty eradication policies have led to a shift in the exogenous and endogenous dynamics of poor areas. In particular, the entry of outside capital has an impact on both efficiency and pathways out of poverty. Poverty eradication policies can bring in or strengthen social capital and play the role of social capital in the eradication of multidimensional poverty among farm households [9]. Clarifying the relationship between social capital and multidimensional poverty is of practical significance for identifying multidimensional poverty in rural households, preventing a return to poverty, and enhancing the sustainable development of individuals or households.

From the history of poverty research, human understanding of the connotation of poverty has gone through a continuous expansion stage from the initial hunger and malnutrition, to income poverty, and then to capability poverty, power poverty, and social exclusion. Measures of poverty have evolved with the perception of poverty [10,11]. The United Nations Development Program released the Human Poverty Index (HPI) in 1997, which measures poverty in monetary terms by using a certain income level as a monetary poverty threshold. Later, in conjunction with the Human Development Index (HDI), the multidimensional poverty index (MPI) was proposed to reflect the characteristics of multidimensional poverty occurrence in terms of monetary and nonmonetary dimensions [12]. Alkire and Foster proposed the “Alkire–Foster method” to calculate the multidimensional poverty index (MPI), which was widely accepted [13,14] and has been widely used to identify and measure multidimensional poverty [15]. Notably, not all studies have followed exactly the same indicators, and in practice, scholars have added or subtracted appropriate adjustments according to data availability and applicability [16,17]. The measurement of multidimensional poverty is the basis of research, and the analysis and identification of influencing factors are the prerequisites for governing multidimensional poverty. Multidimensional poverty was influenced by both intrafamily characteristics and external environment.

A great deal of research has been conducted by scholars on multidimensional poverty [15,18]. Firstly, according to the subject of study, scholars have conducted studies on different subjects such as multidimensional poverty among the elderly [19], female poverty [18], multidimensional poverty among children [20,21], and multidimensional poverty among the overall population in the labor force. Secondly, the factors that contributed to poverty were found to vary. Rural residents tend to have high levels of deprivation in education and basic needs (clean water, sanitation, electricity, and housing conditions), while health, income, social security, and infrastructure development were also negatively associated with the risk of multidimensional poverty occurrence [22,23]. The World Bank, in 2020, reported that multidimensional poverty in rural households was correlated with family size, number of children and education levels while chronic diseases were also able to influence multidimensional poverty status [24]. Moreover, the interaction between multidimensional poverty and other policies has been an important focus and topicality of research. Scholars have conducted a series of studies on the relationship between economic growth [25], income disparity [26], social security, financial services participation patterns [27], public investment in education [28], institutional reform [29], and multidimensional poverty. Particularly, the concept of social capital has received increasing attention. A number of studies have found that social capital plays an important role in reducing poverty [30,31], increasing income [27], and smoothing consumption [32], while the relationship between social capital and multidimensional poverty has received increasing attention from theoretical and empirical researchers. Specifically, disease poverty was a serious problem in rural China, social capital has health-promoting effects [33]. Social capital could moderate the disease poverty relationship [34], and family social capital could influence poor children’s mental health [35,36]. Studies have shown that social capital could promote labor mobility and increase income of farm households, which in turn affects multidimensional poverty of farm households [37]. Education was among the main contributors to becoming poverty [38], social capital has the function of alleviating multidimensional poverty in education poverty [39,40]. The study areas were mainly urban poverty [41], and typical rural areas [1]. As a result, clarifying the relationship between social capital and farmers’ multidimensional poverty is of practical significance for identifying farmers’ multidimensional poverty, achieving effective poverty eradication, preventing poverty return, and enhancing the sustainable development ability of individuals or households.

In conclusion, the existing research combines theoretical-level analysis and empirical research at the realistic level with the characteristics of the study areas, providing academic accumulation for the subsequent social capital in multidimensional poverty governance. However, there is still space for further expansion and the main possible contributions of this paper: (1) in measuring the multidimensional poverty level of farm households, existing studies mainly adopted the MPI index combined with the A-F method. In this paper, the MPI model will be appropriately adjusted to take into account the current policy orientation of poverty alleviation and development in China, and divided into five dimensions, namely health, medical care, education, income and basic living standards. (2) Selecting a representative study area, covering the whole country. The latest Chinese Family Panel Studies (CFPS), which represents the characteristics of rural China, is used for measurement. (3) Theoretical analysis of the inner mechanism of social capital affecting multidimensional poverty in rural households, to make up for the single-dimensional indicators of existing studies that mainly focus on income poverty, while focusing on the impact of social capital on the education, medical, physical and mental health dimensions of multidimensional poverty. Therefore, this paper hopes to make some attempts and breakthroughs in making up for the above three shortcomings. 

There are five parts in this paper, and the remainder of this paper is organized as follows: mechanistic analysis of social capital on multidimensional poverty, date and methods will be presented in Section 2; the findings of the empirical study are given in Section 3; then in Section 4, our discussion will be given; finally, the conclusions and policy recommendations will follow in Section 5.

## 2. Theoretical Mechanisms

Combined with social capital characteristics and with literature, the social capital indicators were decomposed into three dimensions: social network, social trust and social prestige. The mechanisms of action between social capital and multidimensional poverty of farm households based on these three dimensions are mainly reflected in the following aspects.

### 2.1. Social Capital and Rural Household Income

In terms of opportunity mechanisms, the first is that social capital enables poor households to gain access to employment through opportunity mechanisms [42]. This allows them to access labor markets closer to developed regions, which improves their chances of employment [43]. The labor market in developed areas pays more, allowing them to earn higher income and avoid falling into income poverty. Secondly, social capital can change career choice behavior and improve employment quality. Social networks enable the workforce to have more occupational choices prior to employment. In the context of fuller employment opportunities, people can choose to change their behavior and productivity according to their own situation, choosing occupations with higher overall benefits to increase their income. As far as the information mechanism is concerned, most of the poor areas in China are located in information-developed areas, which are prone to fall into the vicious circle of “information asymmetry–low development opportunities–poverty crisis”. The social network, by passing reliable information to each other through friends and relatives to establish an interoperable and shared information platform, enables family labor to obtain higher quality information [9]. The information advantage reduces the sunk cost, opportunity cost and trial and error cost, reducing the probability of falling into multidimensional poverty. Regarding trust mechanisms, social trust plays a role in alleviating multidimensional poverty through two paths: Firstly, by establishing a trust mechanism, it increases information transparency and builds a good resource network between the two sides of the repeated game of employer–employee transactions, which enables rational judgment between the two sides. The second is that job seekers may have access to more information resources through employers, get in touch with more people, expand their social connections, and thus gain access to quality employment opportunities and increase their income.

### 2.2. Social Capital and Rural Family Health

Health manifests itself in two dimensions: mental and physical health. The multifunctional nature of social capital allows it to affect the health of rural families from multiple perspectives. Poor households are often engaged in repetitive types of work that are labor intensive and in harsh working environments, which can impair their health status. Experiencing life crises represented by major and chronic diseases are important causes of multidimensional poverty in rural households [24]. Studies have shown that social capital improves the level of physical and mental health of individuals [33]. The main physical health effect is reflected in access to health benefits [20]. In China, the main medical benefit for poor rural families is new rural cooperative medical care. As an informal system, social capital can optimize the efficiency of medical resource allocation and alleviate the dependence on the market “price” mechanism for medical resource allocation. On the other hand, it can actively promote the improvement and implementation of new rural cooperative medical care with other social medical welfare systems. The scientific and healthy lifestyle can minimize the risk of disease. Social capital has an information dissemination effect, and network members exchange health information with each other to obtain knowledge on physical examination, exercise, prevention, and nutritional acquisition to improve health awareness. The purpose of rural family defense and disease risk reduction is achieved. Individual social capital is often based on blood, geographic and kinship ties, etc., and individual lifestyles are influenced by members of other network ties. At the same time, rural families are able to access medical-related social benefits at a lower cost, reducing the increased costs of trial and error and sunk costs due to unequal information. In terms of mental health, social capital also has the properties of spiritual and cultural resources, which can reduce the psychological stress and negative emotional impact of individuals’ lives. Individuals receive spiritual and respectful support in the group, which compensates for the lack of subjective feelings in work and life and brings psychological satisfaction. The cultural and spiritual spillover effects of social capital provide a path for individuals to develop and shape their spiritual identity and avoid psychological problems that affect their health.

### 2.3. Social Capital and Family Education

Vocational education is among the important paths to interrupt the intergenerational transmission of poverty [44]. Although educational poverty among teenagers in rural households has largely been addressed in China, capability poverty due to vocational education poverty is also an important part of multidimensional poverty [44]. In the labor market, the level of human capital is an essential element for obtaining employment rewards. There is a mutual match between the supply of human capital and job demand, and the education level of workers affects their job type and wage income [28]. Through social capital, individuals and other members develop mutual imitation of skills and accumulation of technical knowledge through cooperative mechanisms to improve their own human capital. This enables them to effectively alleviate employment discrimination and achieve the purpose of fair employment when conducting employment. In addition, the required learning resources can be obtained from social networks through information mechanisms. Through individual initiative, continuous learning and use of online resources to improve their educational attainment and empower the poor with more educational rights, which can improve their human capital better and escape the educational poverty trap. Social capital can enhance individual education value and learning motivation through demonstrative effect, stimulate individual learning motivation, and influence educational attainment. Low productivity levels are among the causes of poverty. Social capital affects educational attainment not only in the labor market, but also in productivity levels. Vocational training opportunities can be accessed through social capital in poor areas in order to influence productivity levels through human capital enhancement. Social capital strengthens the connection between poor areas and the outside world, and facilitates farmers’ access to more and fuller information, including learning vocational skills and understanding industry dynamics. Social capital can change production methods, improve productivity and avoid falling into educational poverty caused by inadequate production information in a timely manner.

## 3. Methods

### 3.1. Study Area and Data

China is the case area for this study. The eastern, western, southern, and northern ends of the Chinese territory are (135°2′30″ E), (73°29′59.79″ E), (3°31′00″ N, 112°17′09″ E), and (53°33′ N, 124°20′ E). China has a land area of approximately 9.6 million square kilometers and a population of 1443.49 million (2020). China is a vast country with a complex and variable climate. Harsh natural conditions, a lack of resources, poor infrastructure, and rapid population growth have led to uneven development between different regions, and the problem of poverty is complex and representative. China is the largest developing country, and its experience in poverty eradication has important implications in the global poverty solution process.

The data in this paper were collected from the CFPS project hosted by the China Social Science Research Center of Peking University. The CFPS covered three levels: individual, family, and community, reflecting the dynamic changes in Chinese society, population, family, economy, education and health, which was typical and representative. The data collection of CFPS adopts the traditional sampling method, and the selected sample covers 25 provinces (municipalities directly under the Central Government and autonomous regions) of China except Hong Kong, Macao, Taiwan, Ningxia, Xinjiang, Tibet, Qinghai, Inner Mongolia and Hainan. The population of the sample area accounts for approximately 95% of the total population of the country, which ensures the representativeness of the sampling results. With a sample size of 16,000 households, the measurement of multidimensional poverty in this paper was analyzed on a household basis, and the CFPS data were able to empirically test the theoretical analysis in this paper to a greater extent.

### 3.2. Measures

Since the introduction of multidimensional poverty, scholars have conducted a series of studies on the measurement of multidimensional poverty and constructed various poverty measurement methods. The most commonly used multidimensional poverty measure is the “Alkire–Foster (A-F) method” a multidimensional poverty analysis method developed by scholars at The Oxford Poverty and Human Development Initiative (OPHI), an initiative of Amartya Sen, in 2007. Specifically, using the A-F method, firstly, the values of each individual on each poverty indicator need to be obtained through a survey, and then a deprivation criterion will be defined for each indicator, according to which each individual is identified as experiencing deprivation on that indicator. Finally, the total number of deprivations for each indicator was summed up and compared with the set threshold value, and deprivation numbers greater than the threshold value were considered to be in multidimensional poverty. It is divided into unidimensional and multidimensional measurements according to dimensions, and the corresponding measurement methods were selected based on practical applications. Since this paper is based on multidimensional poverty research as the departure point, the A-F model multidimensional poverty method was selected for the measurement of poverty index.

(1) Single-dimensional poverty identification. Based on the observed sample n, a matrix X is established, and the number of indicator items is set to p. The matrix X^n,p^ represents the n × p-dimensional matrix, x_ij_ represents the value of household i on dimension j, x_i_ represents the comprehensive value of household i on all indicator dimensions, while x_j_ represents the distribution status of all observed households on indicator j. A threshold poverty criterion s_j_ is established for each indicator to measure whether a household is in poverty for given indicator, meaning that when x_ij_ < s_j_,g_ij_ = 1, others g_ij_ = 0. The deprivation matrix G is obtained by assigning measurements to the sample.

(2) Multidimensional poverty identification. Each indicator of the observed sample is assigned a certain weight w_j_, and the total deprivation status of household i on p indicators is C_i_(k) = w_1_ × g_i1_+ w_2_ × g_i2_ + w_3_ × g_i3_ + … + w_p_ × g_ip_.

Where k represents the threshold value of total deprivation, and since w_1_ + w_2_ + w_3_ + … +w_p_ = 1, the value range of k is [0, 1]. This means that when C_i_ ≥ k, household i is poor and assigned a value of 1; otherwise, it is considered as non-poor and assigned a value of 0. The value of k is adjusted according to the situation. In this paper, we referred to the criterion of k ≥ 30% used by the UN MPI in measuring multidimensional poverty, and based on the consideration that there were 11 indicators chosen to measure multidimensional poverty. To improve and scientific accuracy, those with a poverty index of 33% or more will be identified as multidimensional poverty households.

(3) Multidimensional poverty index calculation. The multidimensional poverty index (*M*) is the multiplication of the poverty incidence and the average deprivation shares. The number of multidimensional poor households and the total deprivation status from the above, the incidence of poverty (H) can be measured. H is the ratio of the number of multidimensional poor households (q) to the number of total observed households in the sample (n), H = q/n. The average deprivation share is the ratio of deprivation value sum of multidimensional poor households to the number of multidimensional poor households in the sample, i.e., A = C_1_(k) + C_2_(k) + C_3_(k) + … + C_n_(k)/q, which leads to M = H∗A = C_1_(k)+ C_2_(k) + C_3_(k)+ … + C_n_(k)/n.

(4) Dimensional decomposition. The multidimensional poverty index can be decomposed based on urban and rural areas, regions, dimensions, etc. The specific decomposition definition varies slightly depending on the actual situation of the problem under study. In this paper, we choose to decompose according to indicators to observe the poverty level of different indicators and their contribution to the multidimensional poverty index. 

### 3.3. Multidimensional Poverty Index Selection, Weight and Depriving Threshold Setting

In terms of indicator selection, regarding multidimensional poverty, this paper was based on the Global-MPI analysis framework, taking into account the actual situation of data acquisition, display fit and other variable additions and deletions. The MPI model was appropriately adjusted by reference researches [8,25] (as shown in Table 1). It was divided into five dimensions, which were health, medical care, education, income, and basic living standards, with the weights still using the equal weight method. 

### 3.4. Variables Selection 

In this study, we selected multidimensional poverty status as the dependent variable (Table 2). Social capital is selected as the core independent variable. Drawing on existing research [8,38,45,46,47], social capital was divided into three dimensions: social network, social trust, and social prestige. We also control for a range of factors that may affect multidimensional poverty status, including individual level, household level, and income structure, were based on the characteristics of the studied households’ poverty. The number of sample observations is 6411. The definitions of variables are shown in Table 2.

### 3.5. Regression Model Setting

The objective of this paper is to investigate the effect of social capital dimensions on multidimensional poverty in rural households. Since multidimensional poverty in rural households is a dichotomous variable taken as yes or no, a binary Logit model was chosen as the model for this paper. The model was as follows: the probability of occurrence of multidimensional poverty in households was assumed to be p, and non-multidimensional poverty was 1 − p. The factors affecting the multidimensional poverty status of households were analyzed by means of a Logit model. The regression equation was set as follows:ln(p/1 − p) = β_0_ + β_1_X_1_ + β_2_X_2_ + β_3_X_3_+ … + β_m_X_m_
where p represents rural multidimensionally poor households and 1 − p represents non-rural multidimensionally poor households. X_m_ denotes the influencing factors of multidimensional poverty of rural households, and m denotes the number of independent variables. β_0_ is a constant term, while β_m_ denotes the regression coefficient of the independent variable, which reflects directions and factors of multidimensional poverty of rural households.

## 4. Results

### 4.1. Multidimensional Poverty Measurement

In order to analyze the multidimensional poverty status of the sample households in detail, the A-F multidimensional poverty index method was applied to measure the multidimensional poverty incidence, multidimensional poverty deprivation share, and multidimensional poverty index. The measured results are shown in Table 3.

The multidimensional poverty index (M) varies with the critical value k, provided that the multidimensional poverty calculation method and the weights of each indicator remain unchanged. Specifically, with higher critical values, deprivation incidence and the multidimensional poverty index are lower. In order to present the corresponding multidimensional poverty indices under different critical values more intuitively, Table 3 shows the corresponding values of multidimensional poverty incidence (H), poverty deprivation share (A) and multidimensional poverty index (M) for the poverty critical value k taken from 0.1 to 0.7. Taking k = 0.3 as an example, from Table 3, H in the sample was 35.36%, A was 38.13%, and M was 13.48%, which means that 35.36% of the sample households were in multidimensional poverty, and poverty households were in poverty at 38.13% of the indicators. As shown in Table 3, H and M vary inversely with k, gradually decreasing as the k increases, but the household poverty deprivation share (A) varies in the same direction as k, gradually increasing as the k increases. That is, the breadth of multidimensional poverty is decreasing but the depth is increasing when the poverty dimension of rural households increases. This means that there were different factors causing poverty among the sample households. In this study, we selected the critical value k = 0.33, identified 1599 multidimensionally poor households and 4812 non-multidimensionally poor households from all the study samples, with the incidence of multidimensional poverty in the sample being 24.94%. From the representativeness of the sample, this basically indicates that multidimensional poverty still exists in rural areas of China, but the probability of occurrence was low, a large proportion of rural households have basically escaped from multidimensional poverty, and sustainability has been improved.

It was concluded above that some rural ’households were still in multidimensional poverty, and clarifying the poverty-causing factors was essential for relevant policy formulation and multidimensional poverty governance. The multidimensional poverty index decomposition reflects the importance of each dimension and indicator. Therefore, we use the multidimensional poverty index decomposition to determine the main poverty-causing factors. As shown in Table 4, the incidence of poverty varies widely across indicators, with the three highest poverty incidence indicators being adult education, health, and chronic diseases in that order, which reached 42.06%, 37.65%, and 29.90%, respectively. This indicates that the three indicators have the highest level of deprivation. They were followed by fuel (25.98%), level of medical care (22.15%) and Engel’s coefficient (22.04%). The level of adult education was the indicator with the highest incidence of poverty in the sample, with 42.06% of the households having adults with less than 6 years of education. This was closely related to the stage of development of education in rural China, where the stage of social development determines the status of individual development. In the stage of underdeveloped society, people do not have access to adequate social welfare and security, highlighted by the low level of education. This was in line with the social development process of the country and was a product of the times. The occurrence of energy and medical poverty are also parts of multidimensional poverty that need to be focused on, and energy poverty incidence is at the forefront of the poverty research field [48].

To further reflect the importance of each dimension and indicator, the contribution of each indicator to multidimensional poverty at the critical value k = 0.33 was calculated in this paper. As shown in Table 4, the contribution rate of each indicator reflects a large variability, with the highest contribution rate being health, followed by adult education status and chronic disease, whose values were 22.49%, 21.20% and 17.74%, respectively. The values all exceeded 15%, reaching a total of 61.43%. This indicates that all three are crucial to the improvement of multidimensional poverty, and that improving the health of household members, education, and the treatment and prevention of chronic diseases can effectively alleviate the existing multidimensional poverty of rural households. The indicator contribution ratios not only reflect the contribution of each indicator to multidimensional poverty, but also reveal the influencing factors behind the multidimensional poverty phenomenon of the sample households. Table 4 showed that the contribution rates of children out of school and electricity poverty were 0.02% and 0.19%, respectively. This indicated that the schooling of school-age children has been properly addressed in recent years, interrupting the intergenerational transmission of poverty due to education. The low poverty contribution rate of electricity, which was a basic living necessity, implied that rural infrastructure development had achieved some success in meeting the basic living needs of rural households. The occurrence of multidimensional poverty due to poor survival at the most basic level was reduced. The above analysis showed that for rural households with multidimensional poverty, health and education were the most important influencing factors for the occurrence of multidimensional poverty. The eradication of multidimensional poverty can be achieved by focusing on improving educational attainment and health status, while requiring that future multidimensional poverty eradication policies be formulated with an emphasis on the importance of education and health.

### 4.2. Logit Regression Results

Based on the above multidimensional poverty measurement results, multidimensional poverty was selected as the explanatory variable, with household multidimensional poverty = 1 and non-poor household = 0. A dichotomous Logit regression model was used to assess the response status of rural household multidimensional poverty to social trust, social prestige, social network, individual characteristics of household head and income from working outside the home and farming.

#### 4.2.1. Effects of Social Capital on Multidimensional Poverty

The results in Table 5 show that spending on the three indicators of the social network dimension significantly reduces the occurrence of multidimensional poverty in rural households in 2018, and all of them are negatively associated with household multidimensional poverty. Specifically, money given by annual relatives in the household has a significant negative effect on rural households’ multidimensional poverty at the 5% level, and both indicators, annual money given to relatives and household expenditure on human gifts, have a significant negative effect on rural households’ multidimensional poverty at the 1% level. The three indicators reflect the state of connection between rural households and their social networks. The impact of social network on the occurrence of multidimensional poverty in rural households is mainly manifested in two aspects: on the one hand, the increase in annual money given to relatives and the expenditure of human gifts is based on the premise that the overall income of the household is good, and the strengthening of social interaction helps the household to improve its ability to obtain external information and reduce the probability of falling into poverty. On the other hand, an active social network helps to increase the household’s resilience to risk. The significant negative correlation between money given by relatives and household multidimensional poverty indicates that relatives are better off and are willing to help the household resist risk and escape multidimensional poverty.

Social trust in neighbors has a significant negative effect on multidimensional poverty in rural households at the 1% level. This indicated that as trust in neighbors increases, the probability of multidimensional poverty in households decreases. Trust in strangers, on the other hand, did not significantly affect the probability of multidimensional poverty in rural households. A main reason may be that trust in neighbors allows for greater opportunity mechanisms that increase the likelihood of cooperation with others and reduce the need to intervene or correct dishonest behavior. Helping each other is an important measure to fight against poverty. Social trust can break the trust barrier of cooperation between people so as to increase the possibility of cooperation. Strictly speaking, individual families are still relatively independent individuals in the overall society, while individual development cannot be separated from the whole. Social trust can further increase employment, income and improve livelihoods based on increased possibilities for cooperation, thereby reducing the incidence of multidimensional poverty among rural households.

In terms of the effect of social prestige on household multidimensional poverty, there was a significant positive association between local kinship and household multidimensional poverty. This suggests that as a household’s local prestige grows, its probability of falling into multidimensional poverty increases. Theoretically, rural families with a higher local social prestige would be more likely to be given preferential access to options, all other things being equal, to improve their chances of working or doing business. As a result, monetary poverty will be improved and multidimensional poverty can be reduced. However, the results did not correspond with theories. There are two possible reasons for this: firstly, with higher reputation in the local community, their sense of responsibility and dedication is higher, and their ideological level determines that they are more able to stand above moral requirements. In this way, they will give priority to other people in greater need when encountering opportunities, thus depriving themselves of jobs or business activities. Secondly, the more social prestige a family has in the local community, the more likely other villagers are to seek help from them when they experience difficulties. Giving financial, material or social help to others may make oneself more vulnerable to multidimensional poverty by reducing their own income and resilience to risk.

#### 4.2.2. Impact of Other Factors on Multidimensional Poverty

The effect of individual characteristics on household multidimensional poverty showed that the age of the household head was significantly and positively associated with the occurrence of multidimensional poverty. This suggested that as the age of the head of household increases, the probability of his or her household falling into multidimensional poverty also increases. The specific reasons are more complicated, but the main reason may be related to the attitude of the household head in identifying opportunities and facing risks. The average age of the household head was approximately 51 years old, which was basically the middle-aged and elderly group. The speed of industrial development and technological progress in the 21st century has far exceeded the learning speed of the groups. Generally speaking, the older a household is, the less able it is to accept new things and the more risk averse it is. The household economy does not grow at a rate that matches the overall development, thereby contributing to the incidence of poverty. 

Specifically, rural household size negatively affects the occurrence of multidimensional poverty in households at the 1% significance level. With a certain number of workers and household income, larger household size means lower per capita income, which makes it easier to fall into income poverty on the one hand, and causes health and education poverty among household members on the other. For rural households, income from working away from home was the main source of household income. With higher income from working, the total household income is higher, and with a constant household size, the per capita income is higher, which can reduce the probability of falling into income poverty. Notably, the effect of farming income on the occurrence of multidimensional poverty in households was not significant. Combined with the overall characteristics of the sample, the per capita ownership of arable land in China is relatively small, only 1.39 acres. Therefore, the contribution of farming income to the overall household income was relatively low and could not help the household out of income poverty. Farming as an occupation was relatively stable and fluctuations between years were within normal limits, which also prevented farming income from being too low to cause income poverty. In conclusion, the effect of farming income on multidimensional poverty was not significant.

## 5. Discussion

### 5.1. Results of Multidimensional Poverty Measurement

Consistent with the results of existing studies, multidimensional poverty existed in China. The incidence of multidimensional poverty was measured to be 24.94% when the poverty threshold k = 0.33 and 35.36% when k = 0.3, while the study by Tang, et al. [49], with a sample size of 5629 at k = 0.3, had a poverty incidence of 29.13%. The discrepancy was due to the inconsistent number of indicators selected. In other studies, 8 indicators were selected to measure multidimensional poverty, while 11 indicators were selected in our study and the weights of each indicator differed slightly, which led to a situation where the results of the study were not the same. The results of [50] showed that the incidence of multidimensional poverty was 10.9% at k = 0.3, which differed significantly from the results of our study. On the one hand, this was caused by the differences in the selection of indicators, and on the other hand, its data processing, which excluded too many data samples and ended up with a sample of 4194, which differed considerably from our study’s sample of 6411, leading to inconsistent results. In the calculation of poverty incidence for each indicator, our results were consistent with previous studies [23,49,50], with adult education being the indicator with the highest incidence of poverty, followed by household members’ health status and clean fuel use. The contribution of each indicator to multidimensional poverty, education and health status together accounted for more than 50%, suggesting that improvements in household educational attainment and health status play an important role in the alleviation of multidimensional poverty.

### 5.2. The Impact of Social Capital on Multidimensional Poverty

Consistent with Gao, et al. [51], this study found that social capital can alleviate multidimensional poverty. The results of this paper also support the findings in previous empirical studies, e.g., Zuo et al., 2018 [52]. That is, there is a link between social capital and poverty. Therefore, improving the level of social capital can help multidimensional poor households to escape from poverty. From the results of this paper, it is clear that increasing trust in neighbors and strengthening ties with members of social networks such as relatives, friends and colleagues can lead to more opportunities and information. It is possible to reduce income and educational poverty among poor households, increase income levels, access to opportunities with options for employment and vocational education, escape from poverty and prevent return to poverty. As an important social relationship, the intensity of interactions with relatives and neighbors affects their poverty status, an observation consistent with previous studies [43]. However, it is social networks that play a major role in social capital during the poverty alleviation process, and our study shows that social trust and social prestige also have a negative and significant effect on multidimensional poverty. As social trust acts on the path of multidimensional poverty, a high level of trust is established among members and thus a relative accumulation of social capital is achieved, and the total amount of social capital significantly suppresses the probability of multidimensional poverty and reduces the degree of poverty among farmers [53]. Social capital mainly enables households to obtain more resources to escape from poverty through social networks, thus achieving poverty alleviation.

### 5.3. Impact of Other Factors on Multidimensional Poverty

Although the obtained poverty incidence is slightly different from the existing studies, the final results found that household size and income status are consistent for multidimensional poverty incidence [2,49], which again proves the findings of the existing studies. The family workforce is an important factor influencing multidimensional poverty. The larger a household is, the smaller its proportion of the number of young adults in the labor force to the total number of household members. On the one hand, it is easy to fall into income poverty, and net household income per capita is an important factor contributing to household poverty. On the other hand, low income can cause health and education poverty among members within the household. The characteristics of household head are significantly and positively associated with multidimensional poverty incidence, and the education of the household head affects farm household poverty [54]. In this paper, the mean age of household heads is approximately 51 years old, basically a middle-aged and elderly group with a generally lower education level. Outworking status significantly and negatively affects the incidence of multidimensional poverty [49], and the results of this paper suggest that working outside the home can reduce the probability of falling into income poverty. It is worth noting that this paper also finds that the effect of farming income on the occurrence of multidimensional household poverty is insignificant. Combined with the overall characteristics of the sample, the occupational attribute of farming determines that it is more stable and its share of total household income decreases as other subsistence income increases.

### 5.4. Implications and Novelty

Theoretical and empirical analysis reveals that social capital not only affects multidimensional poverty in rural China, but also has affected the sustainable development of the destination. First, the highest incidence of poverty, as shown by poor health, adult education and health care, severely affects the sustainability of rural households in destination. The addition of social capital improves these conditions by increasing primary health care inputs, providing adult vocational training, and enhancing basic medical knowledge to improve sustainable development. Second, social capital is closely related to multidimensional poverty governance. Social capital can significantly improve the multidimensional poverty situation of rural households in destinations. Poverty is among the most important factors affecting sustainable development, and the introduction of social capital can help destination families increase their income, escape from poverty, and improve their quality of life, education, and health care to achieve sustainable development in the destination. Similarly, the novelty of the conducted research is reflected in two aspects. On the one hand, it breaks the single-dimensional examination of poverty in rural areas in terms of income. The deeper analysis of poverty in terms of education and living standards provides a more comprehensive theoretical analysis to achieve the post-poverty era to prevent return to poverty, eliminate multidimensional poverty, and promote sustainable development of the region. On the other hand, the introduction of social capital has practical implications for the sustainable development of destinations. Social capital, such as physical capital and human capital, is an indispensable force for social development. Social capital provides rural households with a large number of employment opportunities and development opportunities. By providing opportunities and information, it increases employment and income, improves quality of life, and thus leads to escape from multidimensional poverty and promotes sustainable development.

## 6. Conclusions

Social capital has a significant impact on multidimensional poverty. Using the CFPS data in 2018, this study found that (1) when the threshold value in this paper was chosen as k = 0.33, 1599 multidimensionally poor households and 4812 non-multidimensionally poor households were identified. The multidimensional poverty incidence was 24.94%, and the multidimensional poverty index was 0.1036. In terms of poverty incidence, three of the highest poverty incidence indicators were adult education, health, and chronic disease in that order, reaching 42.06%, 37.65%, and 29.90%, respectively, indicating that these three indicators were the most deprived. For the household multidimensional poor in the sample, health, adult education and chronic diseases contribute significantly to multidimensional poverty, reaching 22.49%, 21.20% and 17.74%, respectively. This indicates that the improvement of family members’ health status and an increase in years of education can alleviate multidimensional poverty. The results of the decomposition of the multidimensional poverty index show that there are large differences in the contribution of different indicator dimensions to multidimensional poverty. (2) Social capital can significantly reduce the probability of multidimensional poverty in rural households. Social network significantly and negatively influenced the occurrence of multidimensional poverty in rural households, and social trust in neighbors had a significant negative effect on multidimensional poverty in rural households at the 1% level. Social prestige is significantly and positively related to rural household multidimensional poverty. (3) Other factors also influence rural household multidimensional poverty, with the age of the household head, household size, and income from working outside the home all significantly associated with rural household multidimensional poverty.

This paper finds that social capital can alleviate rural households’ multidimensional poverty and inhibit a return to poverty. As a social relationship, the intensity of interaction with the outside world affects rural households’ poverty status, and social capital enables rural households to obtain more resources to escape poverty through social networks, and social trust breaks the trust barrier of interpersonal cooperation to escape poverty. Its practical significance is mainly reflected in the implications of social capital in addressing multidimensional poverty for other countries or regions. In this paper, the relationship between social capital and multidimensional poverty is studied in China, the largest developing country with a representative poverty profile. The findings of this study have implications for the alleviation of multidimensional poverty in other regions as well. Specifically, the goal of poverty alleviation through social capital strengthening can be achieved by referring to the findings of this paper and encouraging the participation of social capital in multidimensional poverty governance, with access to employment opportunities and career options for farmers through opportunity mechanisms. Additionally, the information mechanism can be used to establish an interoperable and shared information platform to obtain higher-quality information. Further, the trust mechanism can be used to increase information transparency, establish good resource networks and expand social connection. 

## Figures and Tables

**Table 1 ijerph-20-00217-t001:** Multidimensional poverty indicators and critical deprivation values.

Dimension	Indicator	Description	Weight
Health	Nutrition	Dummy variable, 1 = children with BMI below the thinness threshold, adult members very unhealthy and relatively unhealthy	1/10
	Sickness	Dummy variable, 1 = chronic or sudden illness or multiple illnesses within six months	1/10
Education	Years of education	Dummy variable, 1 = labor force age population and 16–64 years of schooling per capita with primary or less than 6 years of schooling	1/10
	Child out of school	Dummy variable, 1 = at least one school-age child in the household is out of school	1/10
Income	Net income per capita	Dummy variable, 1 = National rural poverty standard in 2015 as the identification standard, below 2800 yuan is judged as poor	1/5
Medical	Rural health insurance	Dummy variable, 1 = none of the household members have rural health insurance	1/10
	Nearby medical level	Dummy variable, 1 = the level of medical care at the location of the visit is very bad or bad	1/10
Basic service	Lighting	Dummy variable, 1 = no electricity in the house, proxy variable electricity = 0	1/20
	Water for cooking	Dummy variable, 1 = use of river and lake water, rainwater, cellar water, pond water, mountain water, etc., for cooking	1/20
	Cooking fuel	Dummy variable, 1 = cooking with non-clean energy sources such as coal and firewood	1/20
	Engel’s coefficient	Dummy variable, 1 = Engel’s coefficient greater than or equal to 60%	1/20

**Table 2 ijerph-20-00217-t002:** Definition of variables.

Primary Indicators	Secondary Indicators	Variables	Definition
Dependent variables	Multidimensional poverty status	Multi-poverty	In a state of multidimensional poverty
Key independent variable	Social trust	strust	Trust in strangers
		htrust	Trust in neighbors
	Social network	fu	Family social interactions, human gift expenses
		fn	Money from relatives
		fp	Money for relatives
	Social prestige	relation	Human Relations
Control variables	Individual level	age	Age of head of household
	Household level	fml_count	number of family members
	Income structure	fl	Income from farming
		fo3	Income from working outside

**Table 3 ijerph-20-00217-t003:** Multidimensional poverty of rural households at different thresholds (%).

Threshold (k)	Incidence (H)	Deprivation (A)	Index (M)	Threshold (k)	Incidence (H)	Deprivation (A)	Index (M)
k = 0.1	88.08	25.85	22.77	k = 0.4	14.97	45.86	6.87
k = 0.2	63.16	31.20	19.71	k = 0.5	4.40	54.72	2.41
k = 0.3	35.36	38.13	13.48	k = 0.6	0.97	63.31	0.61
k = 0.33	24.94	41.52	10.36	k = 0.7	—	—	—

**Table 4 ijerph-20-00217-t004:** Poverty incidence and poverty contribution rate by indicator decomposition.

Indicator	Incidence (%)	Contribution (%)	Indicator	Incidence (%)	Contribution (%)
Income	5.13	10.09	Medical level	22.15	13.28
Health	37.65	22.49	Electricity	1.97	0.19
Chronic	29.90	17.74	Water	4.99	1.48
Dropout	0.02	0.02	Fuel	25.98	6.42
Adult edu	42.06	21.20	Engel coef	22.04	3.23
Med Insure	19.40	10.26			

**Table 5 ijerph-20-00217-t005:** Regression results of the Logit model.

Var	Coef.	Std. Err.
strust	0.00442	0.02308
htrust	−0.0834 ***	0.02224
relation	0.0646 ***	0.02110
fn	−0.336 **	0.14307
fp	−0.394 ***	0.12270
fu	−0.280 ***	0.06671
age	0.00631 *	0.00376
fml_count	0.219 ***	0.02491
fl	−0.0212	0.02225
fo	−0.0949 ***	0.01933

Note: ***, **, and * represent significance levels of 1%, 5%, and 10%, respectively.

## Data Availability

Not applicable.

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
