# Peer review of "The Impact of Social Capital on Multidimensional Poverty of Rural Households in China"

_ijerph, 2022, doi:10.3390/ijerph20010217_

Round 1

Reviewer 1 Report

I read the paper The Impact of Social Capital on Multidimensional Poverty by 2 Rural households in China and found it interesting. The study makes a nice contribution to the subject. The literature is well written.

It can be accepted after small improvements.

-Area - There is no information about the area. You should add a map and some additional data to locate the reader and provide some additional information that helps to understand the area under review.

- Highlight the implications for destination sustainability. Also, the novelty of the conducted study.

-The authors should review the conclusions, thus should present the practical implications of this study and provide more conclusions derived from the results and discussions, respectively present perspectives for further research.

Author Response

Point 1: -Area - There is no information about the area. You should add a map and some additional data to locate the reader and provide some additional information that helps to understand the area under review.

Response 1:  Thank you for your kind comment. Adding a map and some additional data will look more intuitive and more clearly. some additional information that helps to understand the area under review have been given in detail in the paper. Due to the limitations of external factors such as tools and time, we have described the Area part of the paper in more detail, which we hope will be easier to understand.

The added part is as follows:

3.1 Study Area and Data

China is the case area for this study. The eastern, western, southern, and northern ends of the Chinese territory are (135°2′30''E), (73°29'59.79 "E), (3°31'00'N', 112°17'09 "E), (53°33′N, 124°20′E). China has a land area of about 9.6 million square kilometers and a population of 1443.49 million (2020). China is a vast country with a complex and variable climate. Harsh natural conditions, lack of resources, poor infrastructure, and rapid population growth have led to uneven development between different regions, and the problem of poverty is complex and representative. China is the largest developing country, and its experience in poverty eradication has important implications in the global poverty solution process.

The data in this paper were collected from the CFPS project hosted by the China Social Science Research Center of Peking University. The CFPS covered three levels: individual, family, and community, reflecting the dynamic changes of Chinese society, population, family, economy, education and health, which was typical and representative. The data collection of CFPS adopts the traditional sampling method, and the selected sample covers 25 provinces (municipalities directly under the Central Government and autonomous regions) of China except Hong Kong, Macao, Taiwan, Ningxia, Xinjiang, Tibet, Qinghai, Inner Mongolia and Hainan. The population of the sample area accounts for about 95% of the total population of the country, which ensures the rep-resentativeness of the sampling results. With a sample size of 16,000 households, the measurement of multidimensional poverty in this paper was analyzed on a household basis, and the CFPS data were able to empirically test the theoretical analysis in the paper to a greater extent.

Point 2: - Highlight the implications for destination sustainability. Also, the novelty of the conducted study.

Response 2: Thank you for your kind comment. We have added awkward one subsection to discuss the implications for destination sustainability and the novelty of the conducted study in Discussion as suggested.

The added part is as follows:

5.4 Implications and Novelty

Theoretical and empirical analysis reveals that social capital not only affects multidimensional poverty in rural China, but also has affected the sustainable development of the destination. First, the highest incidence of poverty, such as health, adult education and health care, severely affects the sustainability of rural households in destination. The addition of social capital improves these conditions by increasing primary health care inputs, providing adult vocational training, and enhancing basic medical knowledge to improve sustainable development. Second, social capital is closely related to multidimensional poverty governance. Social capital can significantly improve the multidimensional poverty situation of rural households in destinations. Poverty is one of the most important factors affecting sustainable development, and the introduction of social capital can help destination families increase their income, escape from poverty, and improve their quality of life, education, and health care to achieve sustainable development in the destination. Similarly, the novelty of the conducted research is reflected in two aspects. On the one hand, it breaks the single-dimensional examination of poverty in rural areas in terms of income. The deeper analysis of poverty in terms of education and living standards provides a more comprehensive theoretical analysis to achieve the post-poverty era to prevent return to poverty, eliminate multidimensional poverty, and promote sustainable development of the region. On the other hand, the introduction of social capital has practical implications for the sustainable development of destinations. Social capital, like physical capital and human capital, is an indispensable force for social development. Social capital provides rural households with a large number of employment opportunities and development opportunities. By providing opportunities and information, it increases employment and income, improves quality of life, and thus escapes from multidimensional poverty and promotes sustainable development.

Point 3: -The authors should review the conclusions, thus should present the practical implications of this study and provide more conclusions derived from the results and discussions, respectively present perspectives for further research.

Response 3: Thank you for your kind comment. We have reviewed the conclusions, present the practical implications of this study and provide more conclusions derived from the results and discussions, respectively present perspectives for further research as suggested. We added the above needed changes in the Conclusions section.

The added part is as follows:

This paper finds that social capital can alleviate rural households' multidimensional poverty and inhibit return to poverty. As a social relationship, the intensity of interaction with the outside world affects rural households' poverty status, and social capital enables rural households to obtain more resources to escape poverty through social networks, and social trust breaks the trust barrier of interpersonal cooperation to escape poverty. Its practical significance is mainly reflected in the implications of social capital in addressing multidimensional poverty for other countries or regions. In this paper, the relationship between social capital and multidimensional poverty is studied in China, the largest developing country with a representative poverty profile. The findings of this study have implications for the alleviation of multidimensional poverty in other regions as well. Specifically, the goal of poverty alleviation through social capital strengthening can be achieved by referring to the findings of this paper and encouraging the participation of social capital in multidimensional poverty governance. Access to employment opportunities and career options for farmers through opportunity mechanisms. And through the information mechanism to establish an interoperable and shared information platform to obtain higher quality of information. As well as through trust mechanism to increase information transparency, establish good resource network and expand social connection. The above mechanisms enable social capital to play a better role in multidimensional poverty eradication.

Sincerely yours,

Jinfang Wang

Xiaojin Liu

Reviewer 2 Report

I miss more explicit references to Amartya Sen's capability approach and human development in the text and the conclusions. Although the work of Sen is mentioned in the text, there is only one reference to the work of Alkire (2007) as a measurement of poverty index and human development. I think these relationship shoud be stressed more in depth in the paper. 

This paper aims to shows the importance that social capital theory and research along with human development theory and research have for the study of household multidimensional poverty in rural areas. The results obtained are interesting showing the relevance of these approaches for the study of multidimensional poverty. 

Author Response

Thank you for your kind comment. We think your suggestion is reasonable and we have added more details about Amartya Sen's competence approach and human development in the introduction and conclusion as suggested. And we have added some literature as reference 6,7,10,11,13 of the paper.

The revised part is as following:

Poverty is a major priority problem worldwide, and especially for developing countries, poverty eradication is an essential stage in the process of social development. As the largest developing country, China's experience in poverty eradication has important implications in the global poverty solution process. In the past 40 years of reform and opening up, China has made remarkable achievements in poverty eradication [1], and achieved the goal of eradicating absolute poverty by 2020. However, the elimination of absolute poverty does not signify the complete disappearance of poverty in China [2], and the antipoverty efforts in China will transfer to the phase of alleviating relative poverty after the completion of a moderately prosperous society [3,4].Income does not always capture deprivation experienced by individuals [5].The poverty standard measured by income no longer fully and accurately reflects the present situation of poverty in China, and people who have exited poverty may return to poverty because their lack of viable ability to make a basic living leads to the loss of development resources and opportunities. This viable capability included not only an increase in income level, but also a multidimensional capability such as the quality of education, health and living standards [6,7]. Therefore, the quality status of poverty eradication should reflect the multidimensional capability of poverty eradication, shifting from focusing on the quantity of poverty eradication to quality [8], and committing to achieving real poverty eradication. The formulation and implementation of various poverty eradication policies have led to a shift in the exogenous and endogenous dynamics of poor areas. In particular, the entry of outside capital has an impact on both efficiency and pathways out of poverty. Poverty eradication policies can bring in or strengthen social capital and play the role of social capital in the eradication of multidimensional poverty among farm households [9]. Clarifying the relationship between social capital and multidimensional poverty is of practical significance for identifying multidimensional poverty in rural households, preventing return to poverty, and en-hancing the sustainable development of individuals or households.

From the history of poverty research, human understanding of the connotation of poverty has gone through a continuous expansion stage from the initial hunger and malnutrition, to income poverty, and then to capability poverty, power poverty, and social exclusion. Measures of poverty have evolved as the perception of poverty [10,11]. The United Nations Development Program released the Human Poverty Index (HPI) in 1997, which measures poverty in monetary terms by using a certain income level as a monetary poverty threshold. Later, in conjunction with the Human Development Index (HDI), the Multidimensional Poverty Index (MPI) was proposed to reflect the characteristics of multidimensional poverty occurrence in terms of monetary and nonmonetary dimensions[12]. Alkire and Foster proposed the "Alkire-Foster method" to calculate the multidimen-sional poverty index (MPI), which was widely accepted [13,14] and has been widely used to identify and measure multidimensional poverty [15]. Notably, not all studies have followed exactly the same indicators, and in practice, scholars have added or subtracted appropriate adjustments according to data availability and applicability [16,17]. The measurement of multidimensional poverty is the basis of research, and the analysis and identification of influencing factors are the prerequisites for governing multidimensional poverty. Multidimensional poverty was influenced by both intrafamily characteristics and external environment.

[6] Amartya Sen. Commodities and capabilities. Oxford University Press. 1999,104. DOI:10.2307/135247.

[7] Sen AK. Capabilities, Lists, and Public Reason: Continuing the Conversation. Feminist Economics,2004,77-80.

[10] Sabina Alkire; James Foster.Understandings and Misunderstandings of Multidimensional Poverty Measurement. Journal of Economic Inequality,2011,9,289-314.

[11] Alkire, S., Kanagaratnam, U., and Suppa, N. The Global Multidimensional Poverty Index (MPI) 2021. Oxford Poverty and Human Development Initiative, University of Oxford. 2021, OPHI MPI Methodological Note 51.

[13] Sabina Alkire; James Foster. Counting and multidimensional poverty measurement. Journal of Public Economics. 2010,95, 476-487.

Sincerely yours,

Jinfang Wang

Xiaojin Liu

Reviewer 3 Report

The paper is of high quality. It deals with a very complex topic and provides approach to reflect the various factors of rural poverty. It provides broad literature review and the methods applied are appropriate. Overall, the paper deserves publishing after the check of minor spelling or grammatical mistakes.

Author Response

Thank you for your kind comment. We are so happy to hear that. We appreciate the time you have taken to review this article, and all the authors are very motivated by your positive comments. We are eager to make our own contribution to the research topic. Your encouragement means a lot to us and we will continue our efforts. As a result of your comments, we have reviewed the language usage and grammar and made the necessary modifications. Warm greetings and best wishes to you.

Sincerely yours,

Jinfang Wang

Xiaojin Liu